# Nutrition Coverage in Medical Licensing Examinations in Germany: An Analysis of Six Nationwide Exams

**DOI:** 10.3390/nu14245333

**Published:** 2022-12-15

**Authors:** Maximilian Andreas Storz, Alexander Oksche, Ute Schlasius-Ratter, Volker Schillings, Kai Beckschulte, Roman Huber

**Affiliations:** 1Center for Complementary Medicine, Department of Internal Medicine II, Freiburg University Hospital, Faculty of Medicine, University of Freiburg, 79106 Freiburg, Germany; 2IMPP-Institut für Medizinische und Pharmazeutische Prüfungsfragen, 55116 Mainz, Germany; 3Rudolf-Buchheim-Institut of Pharmacology, University of Giessen, 35392 Giessen, Germany

**Keywords:** undergraduate medical education, medical school, human nutrition, nutrition support practices, medical nutrition therapy

## Abstract

The state of nutrition education in medicine is inadequate, with nutrition-related topics being poorly integrated into lectures. Most medical students receive only a few contact hours of nutrition instruction during their entire time at medical school. Identifying potential barriers that may explain the paucity of nutritional knowledge in medical students is thus of paramount importance. The extent of nutrition coverage in the second part of Germany’s nationwide medical licensing exam is currently unknown. We addressed this issue and assessed nutrition content, as well as students’ scores, in this pivotal test prior to their graduation. We performed a post hoc analysis of six nationwide medical licensing examinations (2018–2020) undertaken by 29,849 medical students and screened 1920 multiple-choice questions for nutrition-related content. Nutrition-related questions accounted for a minority of the questions (2.1%, *n* = 40/1920). A considerable number of the questions (*n* = 19) included only a single nutrition-related answer option that was frequently incorrect and served as a distractor. About 0.5% of questions were entirely nutrition related. Despite undeniable barriers, the inclusion of additional nutrition-related examination questions could serve as an incentive to engage students and medical schools in enhancing medical nutrition education. The recently published competence-oriented learning objective catalog in Germany could play a pivotal role in this context, leading to better recognition of nutrition-related topics in medical education.

## 1. Introduction

Nutrition is one of the cornerstones of a healthy lifestyle [1]. Nevertheless, the prevalence of poor dietary intake has increased throughout the 21st century [2] and contributed to an unparalleled increase in non-communicable diseases within the past few decades [3,4]. Globally, approximately 11 million deaths annually are attributable to dietary factors [2,5]. Notably, the World Health Organization (WHO) has estimated that 80% of chronic non-communicable diseases are attributable to modifiable risk factors, including poor diet [6]. Almost one-half of cardio-metabolic deaths in the United States could be prevented through proper nutrition alone [7].

One potential strategy to promote healthier eating patterns is nutrition care provided by healthcare professionals [2]. There is ample evidence that nutrition interventions that promote nutritionally adequate diets can decrease morbidity and mortality [8]. Dietary patterns rich in fruits, vegetables, whole grains, and legumes have been shown to exert beneficial health effects with regard to many chronic conditions [9], including bronchial asthma [10], rheumatoid arthritis [11], and heart failure [12].

There is now a general consensus that physicians should actively engage in providing nutritional advice [13] and must be equipped with a better understanding of the role of nutrition in health and disease [14]. Unfortunately, recent studies have suggested that practicing doctors lack adequate nutrition care knowledge to provide sufficient dietary advice to their patients [8,15,16,17]. Devries et al. pointed out that the “worldwide state of nutrition education in medicine is inadequate” and called for better integration of nutrition-related topics in lectures on disease pathogenesis and treatment [15].

In the United Kingdom, the latest National Health Service (NHS) Long-Term Plan emphasized that “frontline staff need to feel equipped to talk about nutrition” [18]; however, some medical schools offer at most 8 hours of nutrition training [14]. Efforts to promote nutrition education are thus urgently warranted [19]. Calling for action, Adams et al. emphasized that many medical students receive only a few contact hours of nutrition instruction during their entire time at medical school [20].

Although there is still room for improvement, it is of utmost importance to identify other barriers that may explain the paucity of nutritional knowledge in medical students. Identifying these factors could help to enhance nutrition education and support prospective physicians to play a meaningful role in a much-needed dietary shift towards a more sustainable treatment of chronic diseases [9].

One approach to identifying the potentially mediating factors is to look beyond medical school curriculums. Very few studies have assessed the nutritional knowledge required in students’ final-year exams [21,22]. Hark et al. investigated the extent of nutrition coverage in the US Medical Licensing Examinations (USMLE) Steps 1 and 2 in 1986 and 1993. The authors reported that the nutrition coverage in these exams appeared adequate in amount; however, they did not evaluate the content and appropriateness of the examined items [21]. More recent studies are scarce and often limited to specific (local) programs rather than examining national medical licensing examinations. One example is the analysis by Perlstein et al., which assessed the nutrition content of the Bachelor of Medicine/Bachelor of Surgery curriculum at Deakin University Medical School in Australia [22]. The authors found that 10% of the exam questions in the investigated medical course were nutrition-related, yet they were mostly short-answer questions and were asked in the preclinical years of the course.

Comparable data from other countries is scarce but urgently warranted to potentially enhance the nutritional knowledge of medical students. For the very first time, we aimed to assess nutrition coverage in the second part of Germany’s medical licensing exam, a country with numerous medical faculties and a large student body. The main objectives of this study were twofold: (a) to assess nutrition coverage in the second part of Germany’s nationwide medical licensing exam, and (b) to examine how well students scored on the related questions in this pivotal test prior to graduation.

## 2. Materials and Methods

Our study is based on a secondary data analysis of German medical licensing examination questions [23]. The current “Regulation of the Licensing of Doctors” (*Approbationsordnung für Ärzte*) in Germany includes a written medical licensing examination for prospective physicians [24]. This exam is divided into two parts: the first part (also called *Erster Abschnitt der Ärztlichen Prüfung*) and the second part (termed *Zweiter Abschnitt der Ärztlichen Prüfung*) [25,26]. The second part covers the entire spectrum of the clinical sciences and consists of a written test and a combined practical and oral exam.

The exams take place twice annually in spring and autumn. One exam contains 320 multiple-choice questions with 5 possible answer options [25]. The administration and content of the included multiple-choice questions are organized on a nationwide level by the Institute for Medical and Pharmaceutical Examination Questions (*Institut für Medizinische und Pharmazeutische Prüfungsfragen (IMPP)*), located in Mainz, Germany [23,27]. All questions are designed in close collaboration with experts from German medical universities nominated by faculties and/or medical societies.

For this study, we analyzed 6 consecutive second-part medical licensing examinations (autumn and spring 2018–2020). Our analysis included 1920 multiple-choice questions in total. In the first step, we identified all multiple-choice questions with at least one nutrition-related answer option. For this, 3 independent reviewers (Maximilian Andreas Storz, Kai Beckschulte, Alexander Oksche) manually checked all available questions for suitability for inclusion. Table 1 shows the employed inclusion and exclusion criteria. When necessary, we resolved disagreements by including an additional reviewer.

Each multiple-choice question had five possible answer options. The majority of questions (>90%) followed the so-called “Apos-design”, with 5 answer options (A through E) but only one correct answer. A minority of questions followed the Aneg-design, where 5 answer options exist for which the incorrect has to be identified. Thus, it often occurred that some answers covered nutrition-related aspects while others did not. In such cases, we included the respective questions if there was at least one nutrition-related answer option, irrespective of whether this answer was correct or not. We also categorized the eligible questions according to the number of answer options with nutrition-related content (1 nutrition-related answer, 2–4 nutrition-related answers, 5 nutrition-related answers).

In the second step, we assigned the questions to one of five pre-defined nutrition-related categories to gain a better understanding of the frequently covered topics in the examined exams. These topics included (1) “causes and treatment of nutrition-related endocrine and metabolic disorders”, (2) “dietary supplements”, (3) “dietary patterns”, (4) “nutrient content and preparation of dietary products”, and (5) “nutritional deficiencies”.

In the third step, we contrasted students’ ratings in the selected nutrition-related questions with the average exam rating. For this, we used nationwide statistics provided by the IMPP (*Hochschulspezifische Itemanalyse*). Moreover, we performed sub-analyses for each of the aforementioned nutrition-related categories. Finally, we stratified the results by gender (female and male).

All descriptive statistical analyses were performed using STATA 14 statistical software (StataCorp. 2015. Stata Statistical Software: Release 14. StataCorp LP, College Station, TX, USA).

## 3. Results

We identified 1920 potentially eligible multiple-choice questions that were included in the 2018–2020 German medical licensing examinations. All questions were manually screened by 3 independent reviewers and a total of 1880 questions were excluded for not meeting the pre-specified inclusion criteria. Forty questions remained eligible for the final analysis (accounting for approximately 2.1% of all available questions) (Figure 1).

Table 2 shows the number of participants in each exam, stratified by students’ designated periods of study. Our analysis was based on the exam results of 29,849 students of whom approximately 61% were female and 39% were male.

The question classifications based on the pre-defined nutrition-related categories are shown in Figure 1. Almost 63% (*n* = 25/40) of questions covered “causes and treatment of nutrition-related endocrine and metabolic disorders”, followed by questions on “dietary supplements” (12.5% of total questions (*n* = 5/40)). The remaining categories each accounted for ≤10% of the total questions.

The number of nutrition-related questions per exam is shown in Figure 2. It varied substantially across the examinations and ranged from *n* = 4 in autumn 2019 and 2020 (1.3% of total questions) to *n* = 11 in spring 2020 (3.4% of total questions).

Only 22.5% (*n* = 9/40) of the included multiple-choice questions had five nutrition-related answer options. Almost half of those questions (*n* = 19/40) included only a single nutrition-related answer option. In 50% of cases, these answer options were incorrect (so-called “distractors”). The remaining 30% of questions (*n* = 12/40) had more than one but less than five possible nutrition-related answers.

Table 3 shows students’ average results for the included nutrition-related questions stratified by exam. The results were then contrasted with the average percentage of correct answers in the remainder of the exam.

In all but one exam (spring 2018), students performed better in the nutrition-related questions compared to the rest of the exam. The average percentage of correct answers to the included nutrition-related question exceeded the 80% threshold in all but two exams (autumn 2020 and spring 2018). The best performance was found in the autumn 2019 exam, where more than 92.5% of students answered the included nutrition-related questions correctly. The worst performance was found in the spring 2018 exam, where less than 65% of students gave correct answers to the included nutrition-related questions. Given the low number of overall questions (*n* = 40), great caution is warranted in such comparisons as they are susceptible to bias.

We observed a large heterogeneity with regard to the content of the questions. The questions covered a wide array of topics including (but not limited to) vegetarian and vegan diets, ketogenic diets, gluten-free diets, high-fiber diets, high-fat diets, and histamine-free diets. The questions on supplements and nutritional deficiencies covered Vitamin B12 supplementation, iodine supplementation, Vitamin C supplementation, and Vitamin D supplementation. In light of the low number of total questions and the large heterogeneity in the content, we refrained from additional analyses stratified by question category.

Finally, we observed that students scored far above average on all questions focusing on “dietary supplements”. On average, students answered these questions correctly in 94.08% of cases. When looking at all nutrition-related questions (2018–2020), this phenomenon was less pronounced. The average percentage of correct answers, in this case, was 81.63%. Notably, this percentage still exceeded the average percentage of correct answers in the remaining exam questions (75.40%, 2018–2020).

When we analyzed the questions with five nutrition-related answer options only (*n* = 9), the average percentage of correct answers dropped to 76.78%. An analysis stratified by sex (female vs. male participants) revealed no substantial intergroup differences.

## 4. Discussion

The present study assessed nutrition coverage in the German written medical licensing examinations of 2018–2020. By analyzing 1920 questions, we investigated the key characteristics of said questions and examined how well students scored on this pivotal test prior to their graduation.

Despite the large heterogeneity in the content of the identified questions, we observed several trends worth discussing. Nutrition-related questions accounted for a minority of the total questions (2.08%, *n* = 40/1920). A substantial number of questions included only a single nutrition-related answer option, which was frequently incorrect and served as a distractor. Less than *n* = 10 questions were entirely nutrition-related with five nutrition-related answer options. Students scored better on nutrition-related questions compared to the remainder of the exam. Students scored particularly well on questions about dietary supplements. The number of nutrition-related questions did not increase consistently over the investigated 3-year period.

The low number of entirely nutrition-related questions (*n* = 9) did not allow for valid statistical calculations. Hence, we opted for a descriptive analysis. As discussed earlier, there is a concerning gap between the nutrition competencies required for doctors to provide effective nutrition care and the nutrition education provided in medical schools [28]. In this context, the overall number of identified nutrition-related questions appears to be very low in our sample. Our findings are essentially in line with previous studies from the United States of America and Australia, although the number of total questions in our analysis seems to be slightly lower [21,22]. Perlstein et al. assessed the nutrition content of the Bachelor of Medicine/Bachelor of Surgery curriculum at Deakin University Medical School in Australia between 2013 and 2016 [22]. The authors reported that approximately 10% of the exam questions in the investigated medical course were nutrition-related. Nutrition-related questions were mostly short-answer questions and were asked in the preclinical years of the course. A comparison with our study is thus difficult since we excluded typical pre-clinical course questions targeting structural chemical formulas.

Moreover, as opposed to Perlstein et al., we investigated nutrition coverage in a nationwide medical licensing exam with other characteristics and other (Germany-based) legislative requirements. A comparison with the aforementioned study by Hark et al. thus appears to be more appropriate [21]. The authors investigated the extent of nutrition coverage in the US Medical Licensing Examinations (USMLE) Steps 1 and 2 in 1986 and 1993. According to their findings, approximately 12% of the items were nutrition-related in the 1993 USMLE Step 2 exam. The prevalence of nutrition-related items in our study pales in comparison to that number. Then again, it must be mentioned that the study by Hark et al. dates back more than three decades and newer studies are unavailable to the best of our knowledge. A national comparison to other German studies is not possible since we are the first group to investigate the coverage of nutrition-related items in the second part of Germany’s nationwide medical licensing exam.

Internationally, nutrition remains inadequately represented in accreditation and curriculum guidance for medical education at all levels [29]. The fact that nutrition-related items accounted for such a low number of total questions in our study indirectly confirms this trend. Then again, it must be mentioned that the second part of the medical licensing exam in Germany has to cover all clinical disciplines with just 320 multiple-choice questions, leading to an intrinsic challenge. A discussion about the appropriate number of nutrition-related questions is thus controversial.

Furthermore, it may be difficult to construct clear and unambiguous questions in this field that are not open to interpretation from a juridical perspective. The field of nutrition is full of controversies as reflected in the recent diverging (or even contradictory) reports concerning the impact of certain food additives, foods, and nutrients on health that are published each year [30]. This may pose substantial difficulties for multiple-choice question designers. Questions must be designed in a way that the correct (or wrong) answer must be clearly verifiable based on standard textbooks written for medical students at the undergraduate level. Controversial subjects or nutritional research findings that may not be applicable to different populations may thus not be included in questions at all.

Students’ high scoring rates in the nutrition-related questions in our study are difficult to interpret, particularly with regard to the fact that only *n* = 9 questions were entirely nutrition-related. When restricting the analysis to these questions, students’ overall scoring was still slightly better compared to the overall exam scoring. This could be interpreted in multiple ways, e.g., as a high interest in nutrition-related topics, profound knowledge in this area, or simply easy-to-answer questions (which are hard to assess in retrospective analyses). Again, it must be mentioned that the limited number of questions might adversely affect the validity of our results, and caution is warranted when discussing our results.

As discussed earlier, the small number of entirely nutrition-related questions shows that this topic only plays a minor role in German licensing exams. Whether this is justified with regard to the potential of medical nutrition education to reduce the health and economic burden of chronic diseases is subject to debate [31]. Challenges in designing adequate questions certainly play an important role here. It is also impossible to draw valid conclusions from exam results about the quality of medical nutrition education in German medical schools.

A unique chance to improve nutrition-related education in German medical schools is the new national competency-based catalog of learning objectives for medicine (*Nationaler Kompetenzbasierter Lernzielkatalog Medizin 2.0-NKLM*) [32]. This catalog of learning objectives defines competencies that are based on the professional profiles of doctors and should be available upon completion of the German medical degree. In addition to knowledge and skills, the NKLM also includes higher-level learning objectives, such as attitudes, scientific skills, and soft skills. The included learning objectives are readily considered in the first competency-oriented objective catalog (*kompetenzorientierten Gegenstandskatalog Medizin*), which was published by the IMPP in December 2020 and is of paramount importance for licensing exam questions [33]. Updated in May 2021 to fully represent constructive alignment with the NKLM 2.0, the catalog now contains the term “nutrition” almost 500 times.

The latter is an important step forward in the better recognition of nutrition-related topics in medical education in Germany.

Previous studies have demonstrated that nutrition education in residency predicts the frequency of residents’ dietary counseling practices [34]. This applies to both the amount of education and the number of instructional methods used. It is not inconceivable that the same also applies to medical students. Including more nutrition-related questions in the final exams could be an incentive for engaging students and medical schools in enhancing medical nutrition education. Beyond this, other steps are urgently required to enhance medical nutrition education in Germany. New strategies beyond counting dedicated teaching hours and mapping nutrition content are required, aiming at a more contextual understanding of the situated learning that occurs for medical students [35]. Attributing a higher value to nutrition in the medical management of patients may be a key step in this process.

### Strengths and Limitations

The present analysis has several strengths and weaknesses worth mentioning. As for the strengths, we present the results from a nationally representative and centralized licensing examination in Germany. As opposed to other studies, we did not ask students directly for their personal (subjective) opinion on nutritional education in Germany. Instead of relying on bias-susceptible questionnaires, we analyzed standardized data from a large central database comprising answers from 29,849 students who took the exam in the aforementioned time frame. This concept is probably the biggest strength and innovation of our study, particularly with regard to the high number of screened questions (almost 2000 multiple-choice questions). The manual question screening process conducted by three independent reviewers is an additional strength and reduced the likelihood of missing relevant questions. Additional strengths include the stratified analyses by sex and the comparison to the general exam results.

However, the present analysis also has limitations worth discussing. The major limitation is probably the modest number of identified nutrition-related questions (*n* = 40), which did not allow for a reliable statistical sub-analysis stratified by question category. Adding other exams would have been necessary to increase the number of questions eligible for analysis. Yet, we focused on the “current” state of nutritional education in medicine in Germany, and thus refrained from adding older exam cycles (2017 and older). Despite the modest number of eligible questions, we believe that the present work allows for important insights into the topic under discussion and is—to the best of our knowledge—the first German-based analysis in the literature.

## 5. Conclusions

We explored the role of nutrition-related questions in the second part of the written medical licensing examination in Germany. Quantitatively, nutrition-related questions played a minor role in all exams and accounted for approximately 2% of the total questions. Approximately 0.5% of the questions were fully nutrition related.

Although there are undeniable barriers to including additional nutrition-related questions, this could be an incentive for engaging students and medical schools in enhancing medical nutrition education. As suggested previously, this step should extend beyond counting dedicated teaching hours and mapping nutrition content to a more contextual understanding of the situated learning that occurs for medical students. The new national competency-based catalog of learning objectives for medicine may help to improve nutrition-related education in German medical schools. Attributing a higher value to nutritional education in German medical schools requires a tectonic shift in thinking but will certainly pay off in light of the large potential to reduce the health and economic burden of nutrition-related chronic health conditions.

## Figures and Tables

**Figure 1 nutrients-14-05333-f001:**
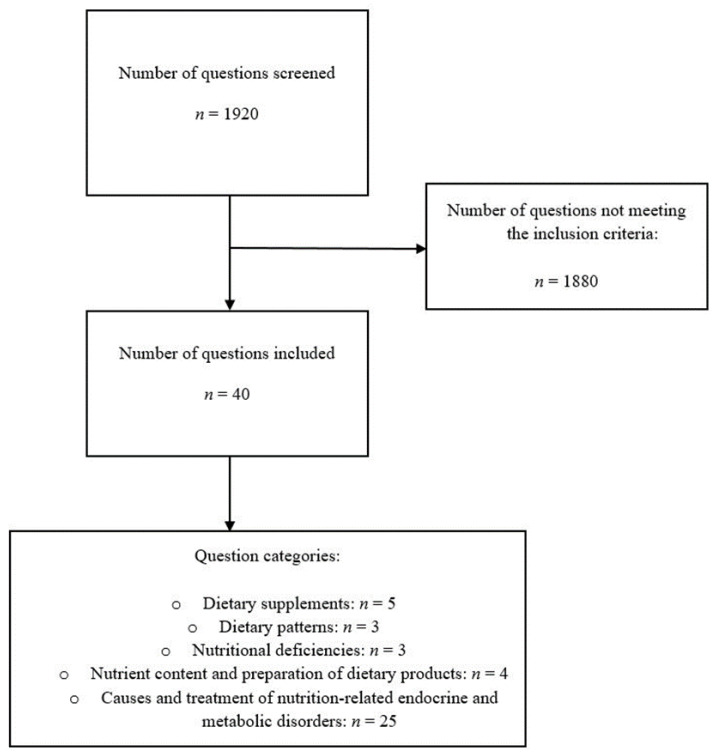
Flowchart for inclusion of multiple-choice questions and question classification.

**Figure 2 nutrients-14-05333-f002:**
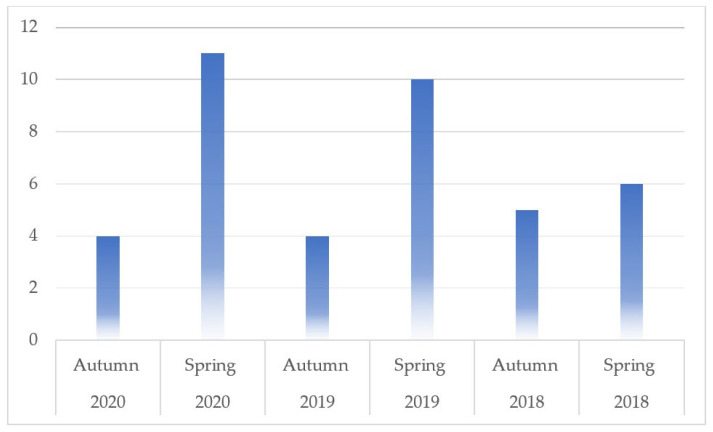
Each bar displays the number of nutrition-related multiple-choice questions in the respective exam.

**Table 1 nutrients-14-05333-t001:** Question inclusion and exclusion criteria. Suitability for inclusion of the screened multiple-choice questions was evaluated based on the displayed in- and exclusion criteria.

Inclusion Criteria	Exclusion Criteria
Questions requiring knowledge about: o Causes and treatments of nutrition-related endocrine and metabolic disorders o Dietary patterns (including gluten-free diets, vegetarian and vegan diets, salt-restrictive diets, ketogenic diets, etc.) o Dietary supplements (vitamins and minerals) o Nutritional deficiencies o Nutrient content and preparation of dietary products	Questions requiring knowledge about: o Eating disorders (including anorexia, bulimia, etc.) o Alcohol-use disorders and alcohol addiction o Herbal medicine and phytotherapy o Structural chemical formulas

**Table 2 nutrients-14-05333-t002:** Distribution of exam participants by gender and designated period of study.

Exam	Female	Male	Female	Male
	Standard period of study (10 semesters)		>10 semesters	
Spring 2018	*n* = 478	*n* = 346	*n* = 2174	*n* = 1308
Autumn 2018	*n* = 2023	*n* = 1331	*n* = 1700	*n* = 1082
Spring 2019	*n* = 492	*n* = 290	*n* = 2231	*n* = 1263
Autumn 2019	*n* = 1788	*n* = 1286	*n* = 1808	*n* = 1112
Spring 2020	*n* = 315	*n* = 239	*n* = 1395	*n* = 733
Autumn 2020	*n* = 1828	*n* = 1272	*n* = 2118	*n* = 1237
Total	*n* = 6924	*n* = 4764	*n* = 11,426	*n* = 6735

**Table 3 nutrients-14-05333-t003:** Percentage of correct answers: nutrition-related questions vs. remaining exam questions.

Exam	Ratio of Nutrition-Related Questions: Remaining Questions	Percentage of Correct Answers: Nutrition-Related Questions	Percentage of Correct Answers: Remainder of Exam	Trend ^1^
Autumn 2020	4:316	75.50%	74.83%	↑
Spring 2020	11:309	89.23%	73.56%	↑
Autumn 2019	4:316	92.55%	73.56%	↑
Spring 2019	10:310	80.81%	74.23%	↑
Autumn 2018	5:315	87.54%	77.60%	↑
Spring 2018	6:314	64.17%	76.77%	↓
Total	40:1880	81.63%	75.40%	↑

^1^ The trend direction (upward pointing arrow with green color or downward point arrow with red color) indicates whether students scored better or worse on the nutrition-related questions in comparison to the remainder of the exam.

## Data Availability

Data associated with this publication will be made available upon reasonable request.

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
