# Peer review of "Nutrition Coverage in Medical Licensing Examinations in Germany: An Analysis of Six Nationwide Exams"

_nutrients, 2022, doi:10.3390/nu14245333_

Round 1

Reviewer 1 Report

-

Author Response

Dear Reviewer,

Please find our comments attached.

Sincerely,

The authors

Reviewer 2 Report

Dear Editor

This article " Nutrition coverage on medical licensing examinations in Germany: An analysis of six nationwide exams” has not a novelty.

Abstract:

·         Line 31: 1920 or 1900. If these questions are fixed in Germany for the field of medicine, it is better to write the same number as 1900.

·         Line 66: National Health Service (NHS) instead of “NHS (National Health Service)”

·         The type of statistical design used in this research should be mentioned.

·         Fig 2: It seems that this form does not provide adequate information to the reader of the article.

·         Discussion text must grammar improve and in some cases it is very weak and maybe there is no discussion at all.

·         Conclusion is very general, try to make it more scientific, comprehensive and concise in detail, especially.

Unfortunately, this article could not explain the importance of the role of nutrition science in medical sciences and instead of scientific discussions, they have relied more on statistics.

Author Response

(The authors gave the same response as above.)

Reviewer 3 Report

The present study sought to investigate the nutritional related questions and examined final year medical exams in Germany. This is an important study and fits with the scope of Journal. However, the following specific points should be addressed before acceptance. 

Abstract

Highlight the important questions

Write the details about the population size

Add some statistical interpretation results

Keywords: Avoid the words used in the title

Introduction

Add the findings of similar studies

Write the novelty of this study before objectives

Materials and methods

Write the details about population size/number of participants

Results and discussion

Enhance the discussion part. 

The questions set (questionnaire asked to the students) may be included in the supplementary material for the reference to the readers. 

Author Response

(The authors gave the same response as above.)

Round 2

Reviewer 2 Report

Dear Editor 

No further suggestion.